# UNDERSTANDING MASKED AUTOENCODERS FROM A LOCAL CONTRASTIVE PERSPECTIVE

## ABSTRACT

Masked AutoEncoder (MAE) has revolutionized the field of self-supervised learning with its simple yet effective masking and reconstruction strategies. However, despite achieving state-of-the-art performance across various downstream vision tasks, the underlying mechanisms that drive MAE's efficacy are less well-explored compared to the canonical contrastive learning paradigm. In this paper, we explore a new perspective to explain what truly contributes to the "*rich hidden representations inside the MAE*". Firstly, concerning MAE's generative pretraining pathway, with a unique encoder-decoder architecture to reconstruct images from aggressive masking, we conduct an in-depth analysis of the decoder's behaviors. We empirically find that MAE's decoder mainly learns local features with a limited receptive field, adhering to the well-known Locality Principle. Building upon this locality assumption, we propose a theoretical framework that reformulates the reconstruction-based MAE into a local region-level contrastive learning form for improved understanding. Furthermore, to substantiate the local contrastive nature of MAE, we introduce a Siamese architecture that combines the essence of MAE and contrastive learning without masking and explicit decoder, which sheds light on a unified and more flexible self-supervised learning framework.

## 1 INTRODUCTION

Recently, self-supervised learning has seen significant progress in the field of computer vision with two dominant paradigms, *i.e.*, Contrastive Learning and Masked Image Modeling. The Contrastive Learning methods (Chen & He, 2021; He et al., 2020; Caron et al., 2020; Dwibedi et al., 2021; Chen et al., 2020; Grill et al., 2020; Chen* et al., 2021; Caron et al., 2021) benefit from learning invariance by contrasting positive and negative image pairs, which are constructed from random data augmentations. On the other hand, the Masked Image Modeling paradigm (Bao et al., 2021; Xie et al., 2022; He et al., 2022; Gao et al., 2022), which is inspired by Masked Language Modeling in the field of Natural Language Processing, involves randomly masking a portion of an input image and learning to reconstruct the missing pixels based on the visible part. Recent studies have shown that the ViT features pretrained with Masked Image Modeling have achieved competitive or even better performance than those with Contrastive Learning, when finetuning on downstream tasks. However, the underlying mechanisms that drive the effectiveness of Masked Image Modeling are still not fully understood compared to the well-explored Contrastive Learning paradigm.

As a typical MIM method, Masked AutoEncoder (MAE) (He et al., 2022) represents a significant milestone for meaningful visual representation learning. MAE paves the way for leveraging the power of masked autoencoding techniques and exploring new possibilities in self-supervised learning. This prompts us to understand how MAE effectively pretrains visual features using a generative learning approach. One crucial aspect of uncovering MAE's underlying mechanism lies in studying its decoder. The key distinction of MAE from previous MIM methods (Bao et al., 2021; Xie et al., 2022; Zhou et al., 2021) is its adoption of an asymmetric encoder-decoder architecture. The encoder is designed to map visible patches only to latent representations, while the decoder reconstructs masked tokens into the original image pixels. Additionally, MAE's decoder demonstrates the remarkable ability to reconstruct images even when subjected to aggressive masking, with a large mask ratio of up to 75%. Given these intriguing yet somewhat ambiguous features, it is essential to gain insight into what MAE truly encodes through a careful analysis of its decoder.

In this paper, we adopt a novel perspective to explain what contributes to "*a rich hidden representation inside the MAE*" (He et al., 2022), focusing on analyzing its decoder's behaviors. Based on the special initialization form of the mask tokens, we first statistically investigate the similarity among all mask tokens' learned attention maps on ImageNet's validation set. The results reveal that the first layer of the decoder primarily relies on the positional information of tokens. While in the subsequent layers, the decoder gradually integrates higher-level semantic information with positional guidance. In another empirical analysis, we conducted a further investigation into the effective receptive field of the decoder. We averaged the attention maps of all mask tokens and observed that the receptive field of the decoder is indeed very limited. This suggests that the decoder primarily relies on local features to perform the reconstruction task. Both of these findings are highly intuitive, as images inherently exhibit locality, and patches in close proximity are often highly dependent. To a significant extent, the training of MAE relies on the image's Locality Principle.

Several recent works (Kong & Zhang, 2023; Zhang et al., 2022a) reconsider MAE in a contrastive learning viewpoint which is indeed a promising direction, as contrastive learning has well-defined formulations and explicit supervision on encoded features. While these methods treat visible patches and masked patches as two views for global contrastive learning, we shed light on explicitly introducing the local receptive field assumption into MAE's masked autoencoding formulation. Our theoretical analysis shows that MAE's reconstruction loss can be interpreted as a region-level contrastive learning loss, with masking as the data augmentation. Moreover, we delve deeper into the role of masking: apart from providing training objectives and data augmentation for MAE, the intensity of masking actually determines the receptive field of the encoder. To further substantiate the local contrastive nature of MAE, we propose a novel self-supervised learning framework, namely Uni-SSL (**Uni**fied **S**elf-**S**upervised **L**earning), which combines the core principles of the MAE and contrastive learning. Uni-SSL adopts a Siamese architecture to perform local contrastive learning between two views augmented by common image augmentations. In contrast to similar works that approximate MAE to contrastive learning (Kong & Zhang, 2023; Zhang et al., 2022a) but still rely on the masking strategy, Uni-SSL offers a significant advantage by removing the dependence on masking. This removal provides more flexibility in network design choices.

Our contributions are as follows:

- We develop a comprehensive understanding framework for MAE, with a novel focus on the decoder. We reveal that the decoder (1) exhibits a transition from positional focus to semantic focus from shallow to deeper layers, (2) reconstructs the image by learning local features within a limited receptive field.

- Based on the assumption of local receptive field, we reformulate MAE as region-level contrastive learning. Moreover, we propose a reasonable framework that can unify MAE and contrastive learning without reliance on masking.

## 2 Understanding MAE as local contrastive learning

In this section, we elucidate that MAE is equivalent to local contrastive learning. We first provide a brief revisit of MAE in Section 2.1 and then investigate the decoding process in Section 2.2. Finally, the region-level contrastive learning form of MAE is proposed in Section 2.3.

### 2.1 A Brief Revisit of MAE

Masked Autoencoders (MAE) (He et al., 2022) is a straightforward yet efficacious self-supervised method for pretraining Vision Transformers (ViT)(Dosovitskiy et al., 2020; Touvron et al., 2021). MAE learns rich hidden representations by masking a portion of the image and then reconstructs the masked patches, leveraging the visible patches.

Formally, given an input image $x$, MAE firstly partitions it into $n$ non-overlapping patches, denoted as $x \in \mathbb{R}^{n \times s}$, where $s$ is the patch size. Then, the $n$ patches are split into two complementary subsets with a random binary mask $m \in \{0, 1\}^n$: the *visible patches* $x^v = x[m]$ and the *masked patches* $x^m = x[1 - m]$. MAE adopts an encoder-decoder architecture. Only the visible patches are fed into the encoder ($f(\cdot)$), which outputs the visible tokens (the representation of visible patches) $z$: $z = f(x^v)$. Then, some learnable mask tokens $M$ are appended to $z$. The visible and mask

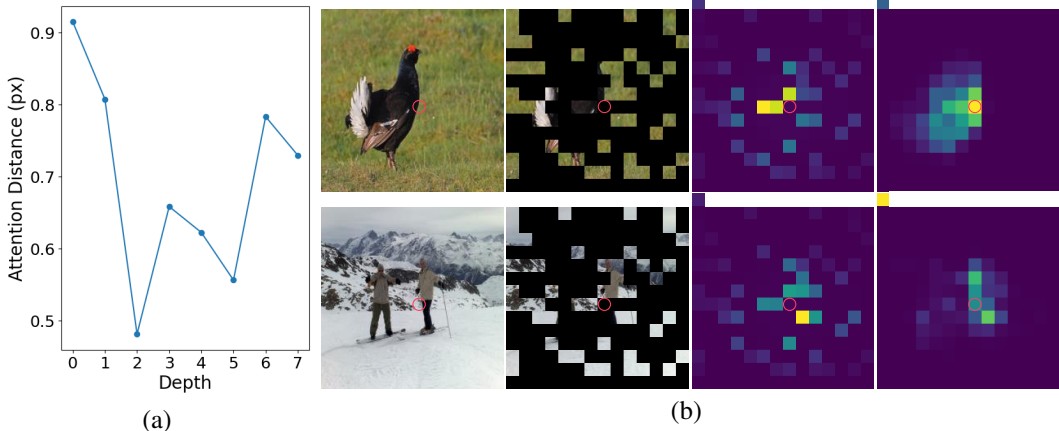

Figure 1: (a) The average attention similarity of each decoder layer. (b) From left to right: the input images, the masked images, the attention maps of the first decoder layer, and the attention maps of the third decoder layer. The red circles (◯) denote the masked tokens serving as queries. The block in the top-left of each attention map is the attention weight for the [CLS] token.

tokens are rearranged back to their original positions in the image. The token sequence is fed into the decoder to reconstruct the original pixels corresponding to the mask tokens. Finally, a simple Mean Squared Error (MSE) loss function is employed for pretraining:

$$\mathcal{L}(x, m) = ||h - x^m||^2, h = g([z, M]),\tag{1}$$

where $g(\cdot)$ denotes the decoder, $[\cdot, \cdot]$ denotes the concatenation of visible and mask tokens based on their positions, and $h$ denotes the predicted pixels of mask tokens.

## 2.2 HOW DOES THE DECODER RECONSTRUCT MASKED PATCHES?

To uncover the inner mechanisms of MAE, it's critical to comprehend the decoder's role in helping the encoder learn "rich hidden representations" in a generative manner, even though the decoder will be discarded after pretraining.

**The decoding process of the decoder.** It is noteworthy that all the mask tokens are initialized from the same mask embedding (denoted as $[MASK]$). Only the added position embeddings $PE$ to the mask tokens are different when fed into the decoder:

$$M_i = [MASK] + PE_i,\tag{2}$$

where $i$ denotes the patch index. Thus, it is evident that the decoded content of different mask tokens is diverse, which implies that MAE's decoding process may mainly be guided by tokens' positional information.

To examine this assumption, we first conduct statistical analysis on the attention maps from different decoder layers using the validation set of ImageNet. To reduce the complexity of the analysis, we deliberately mask all the images with an identical random binary mask (*i.e.*, masked positions are kept the same). Let $\mathcal{I}$ denotes the set of all images, for the $l$-*th* decoder layer, we extract the attention maps from all masked positions of the $i$-*th* image, defined as $A_{l,i} \in \mathbb{R}^{k \times h \times n}$ with $k$, $h$, and $n$ denote the number of mask tokens, the number of heads, and the number of patches, respectively. Then we compute the cosine similarity of each attention map pair $cos(A_{l,i}, A_{l,j})$ and average the similarity scores across the whole image set:

$$S_l = \frac{\sum_{i \neq j}^{|\mathcal{I}|} cos(A_{l,i}, A_{l,j})}{|\mathcal{I}|(|\mathcal{I}| - 1)},\tag{3}$$

where $|\mathcal{I}|$ is the number of images, $S_l$ is the average similarity of the $l$-*th* decoder layer. The higher similarity means the decoder layer relies more on invariant features, *e.g.*, the positional information.

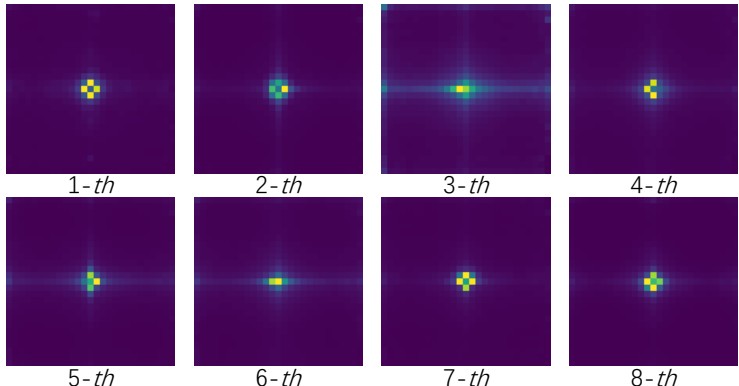

Figure 2: The average relative attention maps for mask tokens of each layer in the decoder.

Conversely, the lower similarity means more reliance on the image-specific information, *e.g.*, the semantic information.

As shown in Figure 1 (a), the average attention map similarity across all images is the highest in the first decoder layer (up to 0.9). Then, the average similarity diminishes in the successive layers. This suggests that different layers leverage distinct features: the first layer is more interested in features shared across images, while successive layers focus more on image-specific features.

To further clarify the decoding mechanism of MAE's decoder, we visualize the attention maps of the first and third layers in Figure 1 (b). In the first layer, where the average similarity is the highest, the attention maps exhibit very similar patterns for the two very different images. This indicates that the first layer primarily relies on positional information. In the third layer, where the average similarity is the lowest, the attention maps reveal that the mask tokens mainly focus on adjacent foreground objects. This indicates that a layer with low similarity places a greater emphasis on capturing semantic information. Combining with Figure 1 (a), where the average similarity starts to be relatively lower from the second layer, we can infer that a deeper decoder is more advantageous for learning semantic information. This aligns with the results of the ablation study about the decoder's depth conducted in the original MAE paper, which demonstrated that a deeper decoder outperforms a shallower one in linear probing.

**Decoder can be seen as a local feature learner.** In Figure 1(b), another noteworthy observation is that the attention weights of the mask token tend to concentrate more on tokens in the closer region. Hence, we further investigate the receptive field of the mask tokens in the decoder. Specifically, we average the relative attention maps of all mask tokens over the whole dataset for each decoder layer. The visualization of the average relative attention maps is shown in Figure 2. We can observe that, across all decoder layers, mask tokens mainly attend to an extremely local area around themselves, which suggests that the decoder primarily learns local features to perform the reconstruction task.

Motivated by this observation, we design experiments to replace the transformer-based decoder with operations that only have a local receptive field. Our first attempt is to replace the transformer layers of the decoder with a weighted average operation, in which the weights are set as a normalized two-dimensional Gaussian, with $\sigma = 1$ and the size of the receptive field is about $5 \times 5$. Then, we adopted a single-layer MLP to reconstruct the masked patches. We pretrain this **Weighted Average** decoder version and the original **Transformer** decoder version of MAE on ImageNet-1K (Russakovsky et al., 2015) for 100 epochs using the same training strategy. As shown in Table 1, surprisingly, the **Weighted Average** decoder achieved a finetuning accuracy

Table 1: Comparisons of different decoders. As the decoder, single-layer convolution and weighted average exhibit effects akin to the transformer.

| Decoder | FT Acc(%) |
| --- | --- |
| Transformer | 82.9 |
| Weighted Average | 82.5 |
| Conv Layer | 82.9 |

of 82.5%, which is only 0.4% lower than the **Transformer** decoder with much fewer parameters. Furthermore, we employ a convolutional layer with a kernel size of 5 as the decoder. We can see

that this **Conv Layer** decoder achieves the same finetuning accuracy as the original MAE, reaching $82.9\%$. Both the visualization of the average relative attention map and experimental results indicate that MAE's decoder can be regarded as a local feature learner.

## 2.3    The learning objective of MAE implicitly aligns local features

**The region-level contrastive learning form of MAE.** Based on the conclusions drawn in Sec. 2.2, we rethink the formulation of MAE's reconstruction loss by explicitly introducing the local receptive field of the decoder into Eq. 1: for each masked position $i$, the mask token $M_i$ attends to its local surrounding tokens $\mathcal{N}_i(z)$. The reconstruction loss of position $i$ becomes:

$$\mathcal{L}(x,m)_i = ||h_i - x_i^m||^2, h_i = g([\mathcal{N}_i(z), M_i]). \tag{4}$$

In the entire training process of MAE, for image $x$, there exists a position $j$ that is masked in two random masking operations with $m_a$ and $m_b$, the loss of position $j$ with respective to $m_a$ and $m_b$ is:

$$\begin{aligned}
\mathcal{L}(x,m_a)_j &= ||g([\mathcal{N}_j(f(x[m_a])), M_j]) - x_j^m||^2, \\
\mathcal{L}(x,m_b)_j &= ||g([\mathcal{N}_j(f(x[m_b])), M_j]) - x_j^m||^2.
\end{aligned} \tag{5}$$

This training objective encourages that the predicted values at position $j$ *w.r.t.* different masking operations approximate the invariant image patch $x_j$.

Then we define the prediction error of the decoder as $e_{pred}(j|m) = g([\mathcal{N}_j(f(x[m])), M_j]) - x_j$, Eq. 5 can be rewritten in the following equivalent form:

$$\mathcal{L}(x,m_a,m_b)_j = ||(g([\mathcal{N}_j(f(x[m_a])), M_j]) - g([\mathcal{N}_j(f(x[m_b])), M_j])) + e_{pred}(j|m_b)||^2, \quad (6)$$

where the second term is the reconstruction loss of MAE. By viewing masking as data augmentation, the first term can be seen as a contrastive loss, ensuring that features obtained based on different random masks are locally similar.

However, in the implementation of MAE, two masking operations for the same image typically occur in two separate epochs, thus the form of Eq. 6 cannot fully describe MAE. We then demonstrate that the aforementioned conclusion still holds for a single forward iteration. Given a random mask $m$, we define the pixel distance between two masked positions $i$ and $j$ as $e_{pixel}(i,j) = x_i^m - x_j^m$. The MAE loss for position $j$ can be rewritten as:

$$\begin{aligned}
\mathcal{L}(x,m)_j =&||g([\mathcal{N}_j(f(x[m])), M_j]) - x_i + e_{pixel}(i,j)||^2 \\
=&||(g([\mathcal{N}_j(f(x[m])), M_j]) - g([\mathcal{N}_i(f(x[m])), M_i]) + e_{pixel}(i,j)) + e_{pred}(i|m)||^2,
\end{aligned} \tag{7}$$

where the first term requires the difference in predicted values at positions i and j should be equivalent to their pixel distance, thus can be regarded as a contrastive loss with $e_{pixel}(i,j)$ as margin.

Eq. 6 and Eq. 7 both indicate that MAE implicitly aligns local features through a region-level contrastive mechanism, we hereby name them as the region-level contrastive learning form of MAE. From this perspective, the reconstruction loss in Eq. 6 and Eq. 7 serves as a constraint to prevent the training of the contrastive loss collapse into trivial solutions, as it prevents tokens from producing identical prediction. This reformulation aids in a more direct understanding of MAE's learning mechanism, as contrastive learning is relatively easier to interpret.

**The encoder mainly focus on local features.** The region-level contrastive form indicates that MAE actually learns features that are invariant to masking in a local region, implying that the encoder should primarily focus on local features. To verify this, we computed the attention distance of MAE (He et al., 2022), DINO (Caron et al., 2021), and supervised pretrained DeiT (Touvron et al., 2021). Attention distance (Dosovitskiy et al., 2020) is defined as the average distance between the query tokens and key tokens, multiplied by the attention weights. It is conceptually similar to the size of the effective receptive fields in CNNs. As illustrated in Figure 3 (a), the attention distance of MAE is significantly lower than that of the contrastive learning method DINO and supervised pretrained DeiT. Many works (Yuan et al., 2021; Liu et al., 2021; Wu et al., 2021) have demonstrated that incorporating the inductive bias of CNNs into ViT can yield better results. We believe that a smaller receptive field is also one of the reasons for the better finetuning performance of MAE.

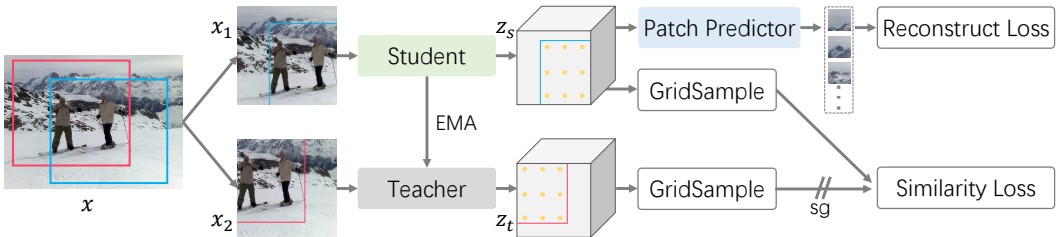

Figure 4: The overall architecture of the proposed Uni-SSL framework. The red and blue boxes indicate the areas of two augmented views, and the yellow dots indicate the sampling points.

**The role of masking.** As the most important hyperparameter of MAE, the mask ratio governs the degree of masking. Intuitively, a smaller mask ratio allows the decoder to find features helpful for reconstruction within a more confined range, thereby indirectly controlling the size of the region for contrastive learning. We pretrain MAE on ImageNet-1K (Russakovsky et al., 2015) for 100 epochs with mask ratios of $0.9$, $0.75$, $0.6$, $0.45$, and $0.3$. The attention distances for different mask ratios are depicted in Figure 3 (b). We can observe that the attention distance under different mask ratios aligns with our intuition: the greater the mask ratio, the larger the attention distance.

We summarize the role of masking into three aspects: 1) As a kind of data augmentation, it enables MAE to learn invariance to occlusion. Since masking operates at the patch level, it results in varying mask strategies across different regions, allowing MAE to perform local contrasts within a single image. 2) It provides training objectives for MAE, and the mask ratio determines the number of training samples. 3) The intensity of masking determines the region's size for contrastive learning, thereby controlling the receptive field of the Encoder. These three roles are not irreplaceable. In the next section, we employ image transformations for data augmentation, utilize contrastive learning approaches to provide training objectives, and constrain the network's receptive field using reconstruction loss for image patches. Through this decoupling, we can design pretraining strategies in a targeted manner.

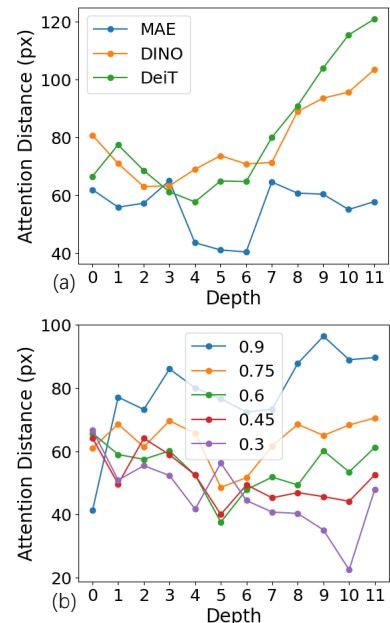

Figure 3: Attention distance for (a) MAE, DINO, DeiT, and (b) MAE with different mask ratios.

## 3 THE UNIFICATION OF MAE AND CONTRASTIVE LEARNING

To validate the aforementioned conclusions, we propose a novel self-supervised framework to unify MAE into the form of contrastive learning, namely Uni-SSL (**Uni**fied **S**elf-**S**upervised **L**earning). Uni-SSL adopts a Siamese architecture, utilizing data augmentation specifically designed for images, and executes local contrastive learning between two views. As a result, masking is not essential for Uni-SSL, allowing it to be compatible with a wider range of network structures and techniques in contrastive learning.

The overall pipeline is shown in Figure 4. Uni-SSL takes two randomly augmented views $x_1$ and $x_2$ of an image $x$ as inputs. The two augmented views are fed into the student network $f_s(\cdot)$ and the teacher network $f_t(\cdot)$, respectively. The student network and the teacher network have identical architecture, which can be either convolutional neural networks (CNNs) or vision transformers (ViTs). The outputs of the student network and teacher network (denoted as $z_s$ and $z_t$, respectively) are dense feature maps, for CNN backbones like ResNet-50, we remove global average pooling, and for ViT backbones, we utilize patch tokens as output.

To acquire positive pairs, we uniformly initialize $K \times K$ sampling points from the overlapping region between $x_1$ and $x_2$, and positive feature pairs are sampled over $z_s$ and $z_t$ according to the mapped sampling points, formally:

$$z'_s = \text{grid\_sample}(z_s, p_1), z'_t = \text{grid\_sample}(z_t, p_2), \tag{8}$$

where $p_1$ and $p_2$ are mapped sampling points, $z'_s \in \mathbb{R}^{(K \times K) \times C}$ with $C$ being the feature channel dimension.

Due to the use of color augmentations such as ColorJitter and RandomGrayscale, directly employing MSE loss in Eq. 6 to minimize the differences between $z'_s$ and $z'_t$ would be inappropriate. Therefore, we employ the 3-layer projection head and cross-entropy loss proposed by DINO (Caron et al., 2021) to ensure the semantic consistency between $z'_s$ and $z'_t$:

$$\mathcal{L}_{sim}(x_1, x_2) = -\text{softmax}(sg(p_t(z'_t))) \cdot \log \text{softmax}(p_s(z'_s)), \tag{9}$$

where $sg(\cdot)$ is the stop-gradient operation, $p_t(\cdot)$ and $p_s(\cdot)$ are projection heads for the student network and teacher network, respectively. The parameters of the teacher network are updated with an exponential moving average (EMA) of the student parameters.

To implement the second term of Eq. 6, which is an image reconstruction loss, we adopt a linear layer as the predictor to estimate raw pixels for each feature vector in $z_s$. We use the unfold operation to extract sliding local patches from image $x$ as the target for the reconstruction loss, formally:

$$\mathcal{L}_{pixel}(x_1) = ||p_{pixel}(z_s) - \text{unfold}(x_1, S)||^2, \tag{10}$$

where $p_{pixel}$ is the patch predictor, and $S$ is the size of local patches. Just as MAE can adjust the size of the effective receptive field by altering the mask ratio, we empirically find that adjusting the local patch size $S$ is a practical way to control the effective receptive field in Uni-SSL. For instance, for a large patch size, the network needs to learn a larger receptive field, whereas, for a smaller patch size, the network only needs to focus on a much smaller region.

The overall objective function of Uni-SSL is calculated as:

$$\mathcal{L}_{Uni\text{-}SSL} = \mathcal{L}_{sim}(x_1, x_2) + \mathcal{L}_{pixel}(x_1). \tag{11}$$

Numerous novel techniques (*e.g.* multi-crop) can be applied to the Uni-SSL framework to achieve better results, but we leave this part for future work.

**Data augmentations.** MAE only employs masking as data augmentation. This is because it uses the input images of the network as targets, thus data augmentations such as ColorJitter are meaningless for MAE.

In order to render Uni-SSL a universal self-supervised learning framework for computer vision, we exclusively employ data augmentation methods designed for images. Specifically, geometric augmentations are RandomResizedCrop with a scale in $[0.25, 1.0]$ and RandomHorizontalFlip with a probability of $0.5$. Color augmentations are ColorJitter and RandomGrayscale with probabilities of $0.8$ and $0.2$, respectively. Blurring augmentation has a Gaussian kernel with std in $[0.1, 2.0]$. For ColorJitter, the strength of brightness, contrast, saturation, and hue are 0.4, 0.4, 0.4, and 0.1, respectively. For RandomResizedCrop, we required the area of the overlapping region to be greater than $30\%$ of the area of the two cropped regions.

**Experimental results and analysis.** Table 2 shows the finetuning accuracy (FT Acc) and linear probing accuracy (Lin. Prob Acc) of ViT-B (Touvron et al., 2021) initialized by different methods.

Table 2: Finetuning accuracy (FT Acc) and linear probing accuracy (Lin. Prob Acc) of ViT-B/16 pretrained by DINO, MAE, and Uni-SSL on ImageNet-1K.

| Pretrain Methods | Epochs | Crops | FT Acc(%) | Lin. Prob Acc (%) |
|---|---|---|---|---|
| Random Init | - | 1 | 78.6 | - |
| DINO | 300 | 12 | 82.8 | 78.2 |
| DINO | 100 | 2 | 81.8 | 68.1 |
| MAE | 100 | 1 | 82.9 | 55.4 |
| Uni-SSL | 100 | 2 | 82.7 | 61.3 |

Table 3: Effect of the number of sampling points $N$.

| K | 3 | 4 | 5 | 6 | 7 | 8 |
|---|---|---|---|---|---|---|
| FT Acc(%) | 82.4 | 82.5 | 82.6 | **82.7** | 82.6 | 82.4 |

Table 4: Effect of the patch size $S$.

| S | 80 | 48 | 16 |
|---|---|---|---|
| FT Acc(%) | **82.6** | 82.1 | 81.7 |

Compared with DINO (Caron et al., 2021) with two global crops, which shares the same network architecture and data augmentation with Uni-SSL, Uni-SSL absolutely improves the finetuning accuracy by $0.9\%$ ($81.8\%$ v.s. $82.7\%$), approaching that of DINO with 12 crops pretrained for 300 epochs. With 100 pretraining epochs, the finetuning accuracy of Uni-SSL is only slightly lower than that of MAE. However, the linear probing accuracy of Uni-SSL is $5.9\%$ higher than MAE. We attribute this improvement to the ability to access the entire image during pretraining.

In Figure 5 (a) and (b), we respectively illustrate the attention distance and the finetuning accuracy curve of Uni-SSL, MAE, and DINO. We can observe that, akin to MAE, the receptive field of Uni-SSL is effectively confined to a very limited range. Figure 5 (b) shows that the finetuning accuracy curve of Uni-SSL remains consistent with that of MAE, manifesting a noticeable difference from DINO. These phenomena indicate that Uni-SSL, through the method of contrastive learning, is able to learn representations analogous to those learned by MAE.

**Ablation studies.** Uni-SSL has two primary hyperparameters: the number of sampling points $N$, and the size of target patches $S$. We first evaluate how the sampling number parameter $N$ affects the finetuning performance. $N$ determines the number of positive pairs involved in the contrastive loss, the larger the $N$, the stronger the constraint on the locality of the network. Table 3 reports the finetuning accuracy with different $N$. It has been shown that as the value of $N$ increases, the finetuning accuracy also exhibits an upward trend. The model performs best when $N = 6$ and the accuracy begins to decline when $N > 6$. The number of tokens output by the backbone network is $14 \times 14$, and a large value of $N$ can lead to overly dense sampling, thereby impairing performance.

As mentioned previously, we utilize the size of target patches $S$ to control the size of the receptive field. In Table 4, we compared the finetuning accuracy for $S$ values of 80, 48, and 16, with $K = 5$. Since we employed a vision transformer with a patch size of 16 as the backbone, these three $S$ values correspond to window sizes of 5, 3, and 1 on the feature map, respectively. It can be observed that different values of $S$ yield significantly varied results, and Uni-SSL performs the best when $S = 80$.

Figure 5: (a) Attention distance and (b) finetuning accuracy curve for Uni-SSL, MAE, and DINO.

**Implementation details.** For pretraining, we conduct experiments on the ImageNet-1K (Russakovsky et al., 2015) training set, with ViT-B/16 (Touvron et al., 2021) employed as the default backbone. For MAE pretraining, the mask ratio is set to 0.75. We use AdamW (Loshchilov & Hutter, 2017) as the optimizer, with a batch size of 1024. The base learning rate $base\_lr$ is initialized with 1.5e-4, and the actual learning rate $lr = base\_lr \times \frac{\text{batch\_size}}{256}$. We adopt a 20-epoch linear warmup, and then the learning rate decays with the cosine scheduler (Loshchilov & Hutter, 2016). For the pretraining of Uni-SSL, we set the base learning rate to 5e-4, with 10 warmup epochs. The projection head used in Uni-SSL has the same setting as DINO (Caron et al., 2021). We also keep the centering operation of DINO and update the center vector with a momentum of $0.9$. The student temperature and teacher temperature are set as $0.1$ and $0.04$, respectively. We set $K = 5$ and $S = 80$ by default except as otherwise noted.

For all the pretrained models, we employed the same finetuning strategy. We use AdamW as the optimizer. The total training epoch number and the batch size are set to 100 and 1024, respectively.

We set the base learning rate as 1.0e-4, and use a 5-epoch warmup. By default, we use the globally pooled patch tokens as inputs for the classifier.

## 4 RELATED WORK

**Contrastive learning.** As the dominant self-supervised representation learning paradigm in the field of computer vision, contrastive learning (Chen & He, 2021; He et al., 2020; Caron et al., 2020; Dwibedi et al., 2021; Grill et al., 2020) learns invariance by comparing random views. A representative work in this domain is SimCLR (Chen et al., 2020), which learns semantic representations by maximizing the similarity between different views derived from the same image within the latent space. MoCo v3 (Chen* et al., 2021) explores the pretraining of vision transformers through the methodology of contrastive learning. DINO (Caron et al., 2021) explores new properties of self-supervised vision transformers. Our work is also related to contrastive learning at the pixel and region levels (Zhang et al., 2022b; Wang et al., 2021; Xie et al., 2021).

**Masked Image Modeling.** In recent years, the development of Vision Transformers (Dosovitskiy et al., 2020; El-Nouby et al., 2021; Touvron et al., 2021) has significantly encouraged the application of Masked Image Modeling (MIM). Originating from Masked Language Modeling, MIM has achieved impressive results in visual self-supervised representation learning. BEiT (Bao et al., 2021) maps image patches into visual tokens using d-VAE (Ramesh et al., 2021) and predicts these visual tokens based on the masked images. SimMIM (Xie et al., 2022) attempts to simplify the algorithmic process of MIM by directly using the original image pixels as the target. MAE (He et al., 2022) employs an encoder-decoder framework to perform image reconstruction tasks. IBOT (Zhou et al., 2021), CAE (Chen et al., 2023), and CMAE (Huang et al., 2022) try to combine contrastive learning and MIM.

**Understanding MAE.** Despite the simplicity and efficacy of MAE, there is a paucity of work dedicated to understanding and analyzing its inner mechanism. Many existing works (Liu et al., 2023; Li et al., 2022; Liu et al., 2022) focus on improving MAE based on intuitive understanding. Cao et al. (2022) primarily focuses on the role of self-attention within the MAE framework. Kong et al. (2023) abstracted MAE as a hierarchical latent variable model, thereby analyzing the mechanism through which MAE learns semantic information. Park et al. (2023) conducted a comparative analysis of the behavioral differences between the MIM and contrastive learning. Kong & Zhang (2023) and Zhang et al. (2022a) reformulate MAE as contrastive learning, sharing similar motivation with us. However, they both consider masked patches and visible patches as two views for global contrastive learning, while we demonstrate that MAE actually conducts contrastive learning between local regions on the masked image.

## 5 CONCLUSION

In this paper, we commence by analyzing the decoding process of MAE and highlight the reliance of the decoder on positional and local information for performing the pixel reconstruction task. By approximating the decoder as a module with a local receptive field, we introduce the region-level contrastive learning formulation of MAE, thereby facilitating a deeper comprehension of its inner mechanisms. Through this reformulation, we uncover that MAE inherently acquires invariance to masking within local regions. We also summarize the roles of masking into three aspects: 1) Serving as data augmentation; 2) Providing training objectives for MAE; 3) Controlling the receptive field of the network. Moreover, to validate our conclusions, we introduce a visual representation learning framework named Uni-SSL, which employs a contrastive learning approach. Experimental results demonstrate that, even without masking, Uni-SSL is still capable of learning representations analogous to MAE, suggesting that Uni-SSL is a feasible way to unify MAE and contrastive learning.

**Limitation.** The major limitation of our work is the implementation of Uni-SSL. Limited by computational resources, we only employ ViT-B/16 as the backbone network and do not experiment with other networks. Additionally, the number of epochs used in our pretraining strategy is much fewer compared to other SSL methods, and the impact of longer training remains unexplored. We will address these two limitations in our open-source code release.

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
