# OpenReview forum: "Understanding Masked Autoencoders From a Local Contrastive Perspective"
_ICLR.cc/2024/Conference — ICLR 2024 Conference Withdrawn Submission_

### Official Review · Reviewer_iWK2 · 2023-10-23

**Soundness:** 2 fair
**Presentation:** 3 good
**Contribution:** 2 fair
**Rating:** 5
**Confidence:** 4

**Summary:**

CL and MIM are the two important self-supervised pre-training methods for computer vision. This paper discusses the connection between them and proposes a new method, which may benefit the community. This paper is well-organized and well-written.

**Strengths:**

The motivation is interesting and the writing is good. The authors provide an interesting finding that MIM and CL methods have a close relationship. It motivates the authors to propose a new method by combining CL and MIM. Moreover, The experiments show the fair performance of the proposed method.

**Weaknesses:**

1. There are currently too few experiments to demonstrate the effectiveness of the approach. At present, the lack of experiments includes longer-epoch training, benchmark object detection, benchmark semantic segmentation, and other experiments.
2. The paper should outperform other contrastive + masked methods, e.g., MST[1], iBoT[2] which were proposed two years ago.

[1] Mst: Masked self-supervised transformer for visual representation. NeurIPS2021.
[2] iBOT: Image BERT Pre-Training with Online Tokenizer. ArXiv2021.

**Questions:**

Please refer to Weaknesses.

---

### Official Review · Reviewer_bqZg · 2023-10-29

**Soundness:** 3 good
**Presentation:** 3 good
**Contribution:** 2 fair
**Rating:** 5
**Confidence:** 4

**Summary:**

This paper empirically analyzes the behaviors of the decoders in MAE. The author finds that 1) the first layer of the decoder primarily relies on the positional information while the subsequent layers obtain higher-level semantic information
2) the receptive field of the decoder is limited. Based on that, this paper reformulates the objective of MAE and proposes a new architecture that combines the spirits of MAE and contrastive learning.

**Strengths:**

1. The writing is well and easy to follow. And the main messages based on the analysis look solid and insightful.
2. The representations learned by Uni-SSL (without masking techniques) are more similar to the mask models instead of contrastive models, which is interesting and verifies the analysis proposed in this paper.

**Weaknesses:**

1. As shown in Figure 3, the attention distance increases with the larger mask ratio. However, both the fine-tuning and linear accuracy are not monotonous in MAE. Is it possible to discuss the trade-off in the choice mask ratio based on the analysis in this paper?
2. In Figure 5(b), the fine-tuning accuracy of DINO is higher than MAE and Uni-SSL while in Table 2 it is the opposite. What differences have I missed?
3. The paper focuses on analyzing the behavior of the decoders in MAE. Is it possible to provide some insights about how to design a better decoder?
4. The connections between the analysis of the decoder, the analysis of the training objective, and the design of the new framework are a little confusing. It would be better to provide a more detailed discussion about that.

**Questions:**

see my comments above.

---

### Official Review · Reviewer_pPsy · 2023-10-31

**Soundness:** 2 fair
**Presentation:** 2 fair
**Contribution:** 2 fair
**Rating:** 3
**Confidence:** 5

**Summary:**

This work has two main contributions: 1)  a comprehensive understanding of the role of the decoder part in MAE, uncovering the fact that the reconstructive decoder part learns local features within a limited receptive field. This work statistically analyzes the similarity among the learned attention maps for all mask tokens on the ImageNet validation set. These findings elucidate that the initial decoder layer predominantly depends on token positional data, whereas in the subsequent layers, the decoder progressively combines more advanced semantic information while maintaining positional guidance. 2) proposing a Siamese architecture combining MIM and contrastive learning in a unified manner.

**Strengths:**

This work has explored the decoder’s role of MAE in helping the encoder learn “rich hidden representations” in a generative manner, uncovering the fact that the decoder part enables to learn local features.

**Weaknesses:**

### Comparison to prior works

This work proposed a combination of Masked Image Modeling (MIM) and contrastive learning (CL) using Siamese architecture, however, this strategy has already been explored in several methods (iBOT [1], CAE [2], and CMAE [3]).  Moreover, this work just commented on these works, not comparing the proposed methods with these prior works. It seems that outstanding points of this work do not exist compared to the prior works.

### Weak experiments

1) very short training epoch: This work was trained for only 100 epochs, which is very short to compare with state-of-the-art methods

2) lacks comparison with prior works (iBOT [1], CAE [2], and CMAE [3]). It would be better to compare with the [MIM+CL] combination methods.

[1] Zhou et al., iBOT: Image BERT Pre-Training with Online Tokenizer, ICLR 2022.
[2] Chen, et al., Context Autoencoder for Self-Supervised Representation Learning, Arxiv, 2023.
[3] Huang et al., Contrastive Masked Autoencoders are Stronger Vision Learners, Arxiv, 2023.

**Questions:**

No questions.

---

### Official Review · Reviewer_zrzk · 2023-11-01

**Soundness:** 2 fair
**Presentation:** 2 fair
**Contribution:** 2 fair
**Rating:** 3
**Confidence:** 4

**Summary:**

This paper discusses MAE is a local-level contrastive learning, and propose a approach to self-supervised learning. This paper is well-written.

**Strengths:**

The authors provide an finding that MAE is local contrastive learning, though some previous paper express the same opinion. Moreover, the paper propose a new method by combining contrastive learning and MAE. Finally, the writing is good.

**Weaknesses:**

1. The paper should cite and compare with related work, like MST, iBoT, CMAE, and so on. They are contrastive + masked methods.


2. The main opinions of this paper is proposed by previous work. Hence, the authors should not ignore them.

3. Current experimental results cannot demonstrate the method is effective. The authors should show detection and segmentation experiments in MAE to fairly compare with MAE and other related work.

**Questions:**

Please refer to Weaknesses.